# Accessing the In Vivo Efficiency of Clinically Isolated Phages against Uropathogenic and Invasive Biofilm-Forming *Escherichia coli* Strains for Phage Therapy

**DOI:** 10.3390/cells12030344

**Published:** 2023-01-17

**Authors:** Swapnil Ganesh Sanmukh, Joana Admella, Laura Moya-Andérico, Tamás Fehér, Betsy Verónica Arévalo-Jaimes, Núria Blanco-Cabra, Eduard Torrents

**Affiliations:** 1Bacterial Infections: Antimicrobial Therapies Group, Institute for Bioengineering of Catalonia (IBEC), The Barcelona Institute of Science and Technology (BIST), Baldiri Reixac, 15-21, 08028 Barcelona, Spain; 2Synthetic and Systems Biology Unit, Institute of Biochemistry, Biological Research Centre of the Eötvös Lóránd Research Network, H-6726 Szeged, Hungary; 3Microbiology Section, Department of Genetics, Microbiology, and Statistics, Faculty of Biology, University of Barcelona, 08028 Barcelona, Spain

**Keywords:** intestinal microflora, inflammatory infections, antibiotic resistance, biofilm-forming potential, bacteriophage

## Abstract

*Escherichia coli* is one of the most common members of the intestinal microbiota. Many of its strains are associated with various inflammatory infections, including urinary or gut infections, especially when displaying antibiotic resistance or in patients with suppressed immune systems. According to recent reports, the biofilm-forming potential of *E. coli* is a crucial factor for its increased resistance against antibiotics. To overcome the limitations of using antibiotics against resistant *E. coli* strains, the world is turning once more towards bacteriophage therapy, which is becoming a promising candidate amongst the current personalized approaches to target different bacterial infections. Although matured and persistent biofilms pose a serious challenge to phage therapy, they can still become an effective alternative to antibiotic treatment. Here, we assess the efficiency of clinically isolated phages in phage therapy against representative clinical uropathogenic and invasive biofilm-forming *E. coli* strains. Our results demonstrate that irrespective of host specificity, bacteriophages producing clear plaques with a high burst size, and exhibiting depolymerizing activity, are good candidates against biofilm-producing *E. coli* pathogens as verified from our in vitro and in vivo experiments using *Galleria mellonella* where survival was significantly increased for phage-therapy-treated larvae.

## 1. Introduction

Approximately 80% of bacterial infections are caused by biofilms, which represents a significant problem for chronic infection treatments due to their reduced antibiotic sensitivity [1,2]. Compared with planktonic cells, biofilms confer an additional mechanism of antibiotic resistance. Therefore, the resistance of biofilms to antibiotics is 100–1000 times greater than that of planktonic bacteria [3]. Moreover, biofilms increase resistance to antimicrobial treatments and disinfectants and protect against environmental stresses [4]. Biofilm-mediated drug/antibiotic resistance is one of the major modes of drug resistance in most biofilm-forming bacteria, including *Escherichia coli* [5]. It is also the way that most biofilm-forming bacteria survive in hostile environments. It has been reported that biofilm development in *E. coli* depends not only on anaerobic conditions but also on its physiological state [6].

It has been observed that biofilm formation is an essential factor responsible for bacterial infections. Certain colonization and virulence factors, such as adhesin (protein), and invasive behavior are critical to the initiation of its synthesis [7]. Biofilms are reported to play a key role in infections associated with enteropathogenic *E. coli* (EPEC), enterohaemorrhagic *E. coli* (EHEC), enterotoxigenic *E. coli* (ETEC), enteroaggregative *E. coli* (EAEC), enteroinvasive *E. coli* (EIEC), diffusely adherent *E. coli* (DAEC), uropathogenic *E. coli* (UPEC), and meningitis and sepsis—meningitis-associated *E. coli* (MNEC) [8,9]. The increased survival rate of UPEC in urinary tract infections is reported in TLR-4 mutant C3H/HeJ mice, lacking an intact innate immune response, following acute infection by the UPEC189 strain due to the formation of intracellular bacterial biofilm-like pods [10,11,12]. The AIEC strain of *E. coli* LF82, associated with Crohn’s disease, demonstrates antibiotic resistance, and can survive and multiply within macrophages [13,14]. Similarly, in vitro studies involving laboratory *E. coli* K-12 reference strain MG1655 demonstrate the interaction of environmental and genetic factors in biofilm formation, which was not restricted to disease-associated clinical isolates [15,16]. The physical protection from the immune system, provided by biofilms, in addition to the ability of numerous strains to develop persister cells that survive very high antibiotic concentrations, are possibly common causes of therapeutic failure seen upon antibiotic treatment [17].

Bacteriophages have been utilized effectively for antibacterial as well as antibiofilm agents, alone or in combination with different antibiotics or antimicrobial compounds/drugs, for in vitro as well as in vivo [18,19,20,21] phage threapies. For example, studies of bacteriophage therapies in BALB/c mice and *Galleria mellonella* larvae have been reported to increase the survival capability in the infected host against different infections for possible human applications [18,19,20,21]. Additionally, phage enzymes, such as endolysins as well as depolymerases, have been explored for such applications [22]. Due to the disadvantages associated with antibiotics and their limitations due to increasing antibiotic resistance development in pathogenic bacterial strains, such as *Pseudomonas aeruginosa*, *Mycobacterium abscessus*, *Staphylococcus aureus*, and many other pathogens, it is very important to identify a suitable alternative against such biofilm-forming pathogens [19,20,21,22].

Community-acquired urinary tract infections (CAUTIs) are the most common infectious diseases affecting more than 150 million people each year globally, where UPEC is considered to be responsible for more than 80% of all CAUTIs [23,24,25]. *E. coli* represents one of the most common members of the normal intestinal microbiota and includes some of the strains involved in a wide range of pathogenic conditions ranging from inflammatory bowel diseases, such as Crohn’s disease or ulcerative colitis [26,27,28], to their prevalence in lungs as an unknown pathogen [29,30,31,32]. Considering the recent report from the World Health Organization (WHO), *E. coli* strains displaying resistance against widely used fluoroquinolone antibiotics are getting more frequent, making their treatment ineffective in most parts of the world [33]. Due to such an urgent emerging situation of antibiotic resistance, alternative treatment options are sought to ensure better disease outcomes, placing phage therapy back into the game.

Biofilms are frequently considered to be a barrier for bacteriophage therapy as the extracellular polymeric matrix produced by some microbial species (i.e., *S. aureus*, *E. coli*, *Klebsiella pneumoniae*, *P. aeruginosa*, etc.) protects these cells from phage infections by forming a barrier. In addition, biofilm aging further impedes phage infection and replication due to the decreasing number of metabolically active bacterial cells [34]. Recently, bacteriophages have been explored against various infectious diseases [17,35,36,37] and seem to be amongst the most promising options for treating urinary tract infections (UTIs). In line with these studies, we are reporting for the first time the in vitro and in vivo efficiency assessment of selected clinically significant phages that harbour depolymerase activity against representative clinical uropathogenic and invasive biofilm-forming *E. coli* strains. We found that irrespective of their host specificity they can be a good alternative for biofilm removal/eradication, which aids the future development of effective phage therapies for UTIs.

## 2. Materials and Methods

### 2.1. Selection and Propagation of E. coli Strains

In this work, we used three different *E. coli* strains, two pathogenic (the uropathogenic CFT073 (ATCC 700928) and the adherent invasive LF82) [38], and the laboratory strain MG1655 (ATCC 700926). All *E. coli* strains were routinely grown in Tryptic soy broth (TSB) and Tryptic soy agar (TSA) media (Scharlab, S.L., Barcelona, Spain). Each strain was selectively grown overnight in TSB at 37 °C in a shaking incubator at 200 rpm. The bacterial cell concentration used for the experiments of phage isolation, in vitro biofilm assay, and one-step growth curve experiments was obtained after adjustment to a final optical density λ = 550 nm (OD_550_) of 0.1, approximately 1 × 10^8^ CFU/mL.

### 2.2. Processing of Clinical Samples for Phage Isolation

The clinical sputum samples from cystic fibrosis patients and urine pooled samples from patients with different, not defined urine infections, were processed before phage isolation. Informed consent was not required because we collected a microbiology laboratory waste product, and all comorbidity data were identified, excluding potential patient risks. Sewage samples were also obtained from the wastewater treatment plant at Prat de Llobregat, Barcelona (Spain), and screened for phage isolation. The Ethic Committee at the Vall d’Hebrón hospital has approved and signed the protocol (PR(AG)275/2019) for the transfer of clinical samples from the hospital to the IBEC lab with all ethical guarantees.

The collected samples were separately incubated in the ratio of 1:5 with the *E. coli* CFT073, LF82, and MG1655 strains in TSB broth at 37 °C overnight. After incubation, the samples were centrifuged separately at 12,000× *g* for 10 min to pellet out all suspended bacterial cell debris. The supernatant lysate was then filtered through a sterile syringe filter with a 0.22 µm pore size to remove any possible contaminant [39].

Isolation of phages against all the *E. coli* strains was carried out in the TSA medium by pour plating as per Lillehaug protocol with some modification [40,41]. For the pour plating assay, 100 µL of bacterial culture at OD_550_ ~0.1 (approximately 1 × 10^8^ CFU/mL) was inoculated with 10 µL of processed clinical lysate separately and incubated at room temperature for 5–10 min before plating. After incubation, the mixture was slightly vortexed and later plated with the help of 1.25% Molten agar at a temperature below 45 °C and incubated at 37 °C overnight for plaque formation. Single plaque showing clear lysis zone and/or double-layered (bull-eyed) morphology were selected. Each plaque was selectively picked and enriched with its respective host, followed by plaque assay. This step was repeated for each isolated phage at least three times until uniformity in plaque formation was observed [42].

### 2.3. Phage Propagation and Lysate Preparation

For phage propagation, 0.5 mL of phage lysate and 5 mL of bacterial host strain at logarithmic phase were added to 5 mL of Trypticase soy broth (TSB) and incubated at 37 °C with agitation (200 rpm) for 18–24 h. At 3 h, 6 h, and 18 h the tube was visually inspected for any bacterial lysis. When bacterial lysis occurred, incubation was stopped, and the sample was centrifuged at 5000× *g* for 20 min at 4 °C. The supernatant was stored at 4 °C.

For laboratory scale propagation, 500 mL of TSB was inoculated with 5 mL of bacterial culture and incubated at 37 °C with constant shaking at 200 rpm. When the culture’s optical density reached OD_550_ = 0.5 ± 0.1 (approximately 1 × 10^8^ CFU/mL), the supernatant obtained above was inoculated at 30 °C at a concentration of 1 × 10^7^ PFU/mL with shaking until considerable lysis (OD_550_~0.2) was observed. Later, 10% *v/v* chloroform was added to the culture with further incubation for more than 30 min at 37 °C, and then the culture was kept at 4 °C without shaking overnight. Finally, the bacterial debris was removed by centrifugation at 5000× *g* for 20 min at 4 °C, and the supernatant obtained was filtered through a 0.22 mm Millipore^TM^ filter and stored at 4 °C as a phage stock [41].

### 2.4. Endotoxin Removal and Bacteriophage Purification

For phage precipitation and enrichment, two different steps of PEG precipitation were performed as previously described [38,40] with some modifications. Phage stocks were centrifuged at 10,000× *g* for 20 min to remove bacterial cells/debris. PEG-8000 (2 g/L) and NaCl (3 g/L) were then added to the resulting supernatant with proper mixing and centrifuged at 10,000× *g* for 10 min at 4 °C. Pellets and cell debris were discarded. Again, the supernatant was added to a solution containing PEG (9 g/L) and NaCl (2.9 g/L) with proper mixing and left at 4 °C overnight (or −20 °C for 1–2 h). The lysate was centrifuged at 10,000× *g* for 20 min at 4 °C to pellet all phage. At this stage, the pellet can be resuspended in 1 mL of 1X PBS buffer. Endotoxin removal was performed using the ToxOut™ Rapid Endotoxin Removal Kit (Thermofisher, Waltham, MA, USA) according to the manufacturer’s protocol. This protocol was repeated several times to keep endotoxin levels below 0.05 EU/mL. Further traces of endotoxin were removed by diluting these purified phages. Purified phages were later analyzed to determine their titer, and purified bacteriophage stocks were further diluted and used for in vivo studies [43,44,45].

### 2.5. Cross Infectivity among Isolated Phages

For the spot test, 200 μL overnight culture of each *E. coli* strain was poured plated with TSA agar and kept for 45–60 min to solidify. After solidifying, 10 μL of each isolated phage with a titer greater than 10^8^ PFU (plaques forming units)/mL was spotted on the plate to check the cross-reactivity. The plates were left to dry and were incubated at 37 °C. The plates were visually inspected for lysis zones [41]. For phage isolation, the spot test was performed in triplicate following their inclusion in the IBEC library. Phages showing consistent lysis events were considered for the experiments, and plaque characteristics, such as clear lysis zone and/or double layer (bull-eyed), latent period, and high burst size, were some of the factors used for phage selection. Special importance was given to the phages exhibiting double layer (bull-eyed) property, which is a representation of phage-bound depolymerase activity.

### 2.6. Phage One-Step Growth Characteristics

The one-step growth curves of the selected phages were performed as described previously, along with some modifications [42]. The different multiplicity of infection (MOI) within the range of 0.1 to 0.0001 were selected to determine the effective phage-host concentration for understanding the latent period and the burst size of phages against their respective hosts. All the experiments were performed in triplicate, and the results are the mean ± standard deviation. The latent period and burst size of these phages was determined by observing the number of phage particles’ changes during the experiments. Based on the number of PFU/mL, the latent period and the burst size were determined by dividing the average PFU/mL of the latent period by the average PFU/mL of the last three time points of the experiment, as reported previously [46,47].

### 2.7. Scanning Electron Microscopy (SEM) for Studying the Lytic Phage Activity

Scanning electron microscopy was performed to observe the phage-mediated bacterial lysis. For sample processing, 50 mL of each *E. coli* strain grown overnight at OD_550_ = 0.1 (approximately 1 × 10^8^ CFU/mL) was inoculated with its respective phage at a minimum concentration of 10^6^ PFU/mL and incubated overnight at 37 °C with constant shaking condition at 200 rpm in 100 mL falcon flasks. Following incubation, the flasks were kept at room temperature to let the lysed bacterial debris settle at the bottom for 20–30 min. The coverslips (Thermofisher, Waltham, MA, USA) of 20 × 20 mm were first sterilized with 70% ethanol and later wiped with milli-Q water with the help of tissue paper. For each phage, 100 µL lysate was equally spread on the coverslip with another thin coverslip slide in one direction to form a homogeneous smear. The coverslips were dried in sterile conditions, preferably in Petri-plates. The coverslips were washed twice by submerging them in 0.1 M PBS pH 7.4 and were fixed with the help of 2.5% glutaraldehyde (Sigma, Spain) diluted in 0.1 M PBS pH 7.4 for 2–3 h. After fixation, the coverslips were again washed 3–4 times with 0.1 M PBS pH 7.4. Dehydration steps were performed by emerging the fixed coverslips in different concentrations of ethanol (Sigma Spain) (from 50%, 60%, 70%, 80%, 90%, and 100%) and were placed over filter paper and dried in a Petri plate upside down before observation with the help of NOVA NanoSEM 230TM through-the-lens detector of the secondary electron (TLD-SE) to obtain images of ultra-high resolution.

### 2.8. Transmission Electron Microscopy (TEM) of Bacteriophages

For sample processing for transmission electron microscopy, 30 µL of purified bacteriophages from samples with a concentration of around 10^8^ PFU/mL in 1x PBS solution were placed on a paraffin tape initially cleaned with 70% ethanol. A carbon-coated copper grid was placed over the sample drop to absorb for 25–30 min and later removed. The excess sample around the grid was released with the help of filter paper without disturbing the mesh. The same side of the grid was placed over the 30 µL of 2% uranyl acetate (Sigma, Spain) staining solution for 30 s, and again the excess solution was removed with the help of filter paper. The grids were placed over filter paper and dried under a Petri plate upside down to avoid contact with the wet surface. They were later observed with the help of J-1010 (Jeol) coupled with a CCD camera Orius (Gatan) with software Digital Micrograph (Gatan) at an 80 kV accelerating voltage.

### 2.9. Static and Continuous Flow Biofilm Assay

For the in vitro static biofilm assay, 200 μL of each bacterial suspension at OD_550_ = 0.1 (approximately 1 × 10^8^ CFU/mL) was inoculated in 96-well polystyrene plates with a flat bottom (Corning 3596 Polystyrene Flat Bottom 96 Well Corning, NY, USA) in triplicates and incubated overnight in TSB medium + glucose 0.5% at 37 °C without shaking. At different time points (24, 48, and 72 h), the media was removed, and wells were washed three times with 1X PBS pH = 7.5 (Fisher Scientific, Madrid, Spain) at 70 rpm for 3 min. After 72 h, formed biofilms were treated with 200 μL of phage sample with a concentration of 1 × 10^6^ PFU/well in triplicate with control and incubated again at 37 °C for 24 h. Here, the phages used for the treatment were diluted in TSB broth, also used in control wells. After treatment, the wells were washed three times with 1x PBS and stained with 200 μL of 0.1% (*w/v*) Crystal violet for 5 min. Then, 200 µL/well of 30% acetic acid (Sigma, Madrid, Spain) was used to elute the Crystal violet, and biomass was determined by measuring the absorbance (OD_570_) using the SPARK Multimode microplate reader (Tecan, Männedorf, Switzerland). The formed biofilms were categorized depending on their biofilm formation as per Stepanović et al. [48].

For the continuous flow biofilm assay, flow-cell chambers were employed according to the protocol previously described [49]. Briefly, the biofilms were grown in Luria Bertani (LB) broth 0.1× enriched with 0.002% glucose and pumped through the flow-cell system using a peristaltic pump (Ismatec ISM 943, Ismatec) at a constant rate of 42 μL/min. An inoculum of 250 μL of the pathogenic CFT073 strain, the best biofilm former, at OD_550_ = 0.1 (approximately 1 × 10^8^ CFU/mL) was introduced to the system, and 2 h of static conditions were provided to favor the attachment step. Afterward, the flow was restored for 96 h, and the mature biofilms were treated for 12 h with 200 μL of tested phages at a concentration of 1 × 10^7^ PFU/mL. LB broth and Ciprofloxacin (CPX) 1 µg/mL treated channel were implemented as negative and positive controls, respectively. Finally, biofilms were stained with the LIVE/DEAD BacLight Bacterial Viability kit (Thermofisher, Waltham, MA, USA) to image with the 20×/0.8 air objective of a Zeiss LSM 800 confocal laser scanning microscope (CLSM). The obtained data were visualized and analyzed by Image J. Biomass (µm^3^/µm^2^) and Thickness (µm) measurements of biofilms were calculated from live cells data (green channel) by COMSTAT2 analysis software

### 2.10. In Vivo Phage Therapy Efficacy in the Galleria Mellonella Infection Model

The *G. mellonella* larvae were maintained and injected, as previously reported by our laboratory [50]. Bacteriophage samples were prepared using ToxOut™ Rapid Endotoxin Removal Kit (Thermofisher, Waltham, MA, USA) to effectively eliminate endotoxins up to <0.05 EU/mL in solutions containing proteins or pharmacologically important components as per the manufacturer’s protocol with some modifications as discussed above. For this experiment, a single *E. coli* strain (CFT073) producing high biofilm and high-biofilm-disrupting phages (IBEC 40 and IBEC 77) were selected for studying in vivo survival in *G. mellonella* according to the in vitro biofilm assays (static and in-flow biofilm assays).

The bacterial culture was centrifuged at 1500× *g* for ten minutes and resuspended with 1X PBS during three washes to remove any possible toxins from the overnight culture. Phage toxicity of endotoxin-free diluted purified phages was first assessed by injecting larvae with 10 μL of different phage concentrations from 10^3^ to 10^5^ PFUs/larva prepared in 1X PBS. All solutions were prepared in 1X PBS. To determine treatment efficiency, larvae were injected with 10 μL of either 1X PBS or the selected *E. coli* CFT073 strain as controls. For the phage treatment group, larvae were injected with 10 μL of *E. coli* CFT073 (10^6^ CFUs/mL). One hour post-infection, purified bacteriophages were injected with 10^4^ PFUs/larva. A second reminder phage dose was also tested five hours post-infection.

After bacterial infection, an antibiotic control group was injected with one dose of ciprofloxacin (CPX) (20 mg/kg larva) [51,52]. Additionally, the mixed combination of ciprofloxacin with each respective phage was tested one hour post-infection. Another group injected with ciprofloxacin at one hour post-infection was followed with the injection of the respective phage 5 h post-infection. All experiments contained 8 larvae per condition. Larvae were incubated at 37 °C throughout the experiments. Following injection, the larvae were frequently observed to survive up to 48 h or 55 h.

### 2.11. Statistical Analysis

The statistical analysis of all the experiments was performed using values expressed as mean ± standard deviation (SD) in GraphPad Prism 9.00 (GraphPad Software, San Diego, CA, USA) software package. All the experiments performed were repeated at least three times. The *p*-values of < 0.05 were considered significant.

## 3. Results

### 3.1. Isolation and Selection of Phages

For the isolation of *E. coli* phages, the clinical (urine and sputum) and sewage sources were tested against the three different *E. coli* host strains (CFT073, LF82, and MG1655) to detect the available phages specific against each host. The detection of phages was dependent upon their availability in these sources and their host specificity; hence, different numbers of phages were selected for each strain from different sources based on the plaque morphology. The source of isolation and plaque characteristics like turbidity, plaque diameter, and double-layered and/or bull-eyed morphologies were considered crucial for phage isolation and selection. All 21 phages were isolated against the three *E. coli* strains. For uropathogenic *E. coli* CFT073, 3 phages, namely, IBEC 77, IBEC 78, and IBEC 98; against the adherent invasive *E. coli* LF82, 5 phages, namely, IBEC 36 through IBEC 40; whereas against the laboratory strain *E. coli* MG1655, 13 phages, namely, IBEC 41 through IBEC 53, were selectively isolated as represented in Table 1. Phage selection was based on plaque morphology involving phages that produce distinct and visually distinguishable plaques when applied to a bacterial lawn. Plaque characteristics includes turbidity (clear or blur), and specific properties like formation of double layered or bull eyed plaques as represented in Table 1.

### 3.2. Spot Test and Cross-Infectivity among Isolated Phages

The spot test was performed to determine the cross-infectivity of 21 isolated phages against non-specific *E. coli* hosts. It was observed that the original host against which the phages were isolated was not as cross-reactive as observed after performing the spot test assay a minimum of three times. Considering the common differences, plaques were first categorized based on their diameter sizes ranging from small (<0.5 cm), medium (0.5–1 cm), and large (>1 cm) and later based on turbidity (from blur to clear). Considering the importance of phage-bound depolymerase enzymes in biofilm removal, plaques showing double-layered or bull-eyed plaque morphology representing depolymerase activity were considered.

The selection of phages for the experiments was based on these three selection criteria; clear plaque morphology, double halo formation (bull-eyed morphology), and/or both, as these properties were reported to be effective in dispersing biofilm and making phage therapy effective as reported in different bacterial pathogens [53,54,55]. As all these features were observed in the clinical phage isolates, IBEC 36, IBEC 40, IBEC 48, and IBEC 77 phages were selected for further in vitro and in vivo studies. The plaque morphology figures are provided in Appendix A.

The sole cross-infective phage was IBEC 40, which was isolated against *E. coli* LF82 and could also infect *E. coli* MG1655 but not the *E. coli* CFT073 strain. Since its plaques showed depolymerase activity and had clear morphology, it was considered an effective candidate phage for further in vivo studies as a neutral (non-infective) phage-nanoparticle-carrying bound depolymerase. The cross-reactivity of all the isolated phages observed from the spot test is provided in Table 2. As the sources of the isolated phages varied, so did their host specificity and plaque morphologies. The plaques showing intermediate lytic activity based on the plaque turbidity or blur morphology were not considered lytic to avoid false positive results and lysogen formation.

### 3.3. Phage One-Step Growth Curve

The one-step growth curve for the IBEC 36, IBEC 48, and IBEC 77 were performed to find out the latent period and average burst size while infecting their original hosts. For IBEC 36 and IBEC 77, a multiplicity of infection (MOI) used to determine their one-step growth curve was 0.0001, whereas for IBEC 48, the MOI was 0.001 due to its long latent period and small burst size. The observed average burst size of IBEC 36 was approximately 250 PFU/cell, IBEC 48 was about 25 PFU/cell, and IBEC 77 was around 150 PFU/cell following primary propagation and enrichment with their original hosts (Figure 1).

### 3.4. Scanning Electron Microscopy Studies

The scanning electron microscopy revealed the lytic activity of IBEC 36, IBEC 48, and IBEC 77 against *E. coli* LF82, *E. coli* MG1655, and *E. coli* CFT073, respectively, as represented in Figure 2. The control for each cell is provided in Appendix A. The specific phages were selective against their hosts, and no-cross reactivity was observed and was confirmed by spot assays.

### 3.5. Transmission Electron Microscopy Studies

Transmission electron microscopy revealed the morphology of the three phages, namely IBEC 36, IBEC 48, and IBEC 77, used in this study (Figure 3). IBEC 36 and IBEC 48 were morphologically similar and belong to the Myoviridae family, whereas IBEC 77 belongs to the Podoviridae family as per preliminary observation through TEM analysis.

### 3.6. In Vitro Biofilm Assay

Next, we evaluated the antibiofilm activity in static conditions of the three selected phages, IBEC 36, IBEC 48, and IBEC 77, which showed promising characteristics based on their plaque morphology and lytic activity depending on their source of isolation as well as depolymerase production capacity against *E. coli* strains LF82, MG1655, and CFT073, respectively. The classification of in vitro biofilm formation after 72 h of incubation was carried out as per Stepanović et al. (2000) and, as expected, it was observed that *E. coli* CFT073 forms a more robust biofilm in vitro followed by *E. coli* MG1655, which forms moderate, and *E. coli* LF82, which produces weaker biofilms [48].

As seen in Figure 4, the IBEC 36, IBEC 48, and IBEC 77 phages significantly reduced the biofilm formed after 72 h of incubation. However, in the case of *E. coli* CFT073, a considerable reduction in the biofilm formed was observed (68%) as compared to the *E. coli* LF82 (47%) and *E. coli* MG1655 strains (40%). As IBEC 77 is a depolymerase-enzyme-producing phage, these results suggest the cumulative effectiveness of phages and their enzymes against the biofilm produced by uropathogenic *E. coli* CFT073 strain. The effective minimum concentration used to obtain the significant reduction in all three strains was 10^6^ PFU/mL for all the three phages after 24 h of incubation (*p* < 0.0001) (Figure 2).

Then, we selected the *E. coli* strain CFT073 due to its characteristic of high biofilm production and two phages (IBEC 40 and IBEC 77). IBEC 40 phage was randomly selected to verify if the non-specific effect of phage-bound lytic enzymes, such as depolymerases, have antibiofilm effects (in vitro) and/or enhance bacterial survival in vivo. The effects of the phages IBEC 40 and IBEC 77 were tested against biofilms formed under flow conditions because of the better representation of a real infection obtained in this model in comparison with microtiter plates. In concordance with our previous results, IBEC 77 phage significantly reduced the thickness and the biomass of *E. coli* strain CFT073 dynamic biofilms (Figure 5). However, since IBEC 40 is not cross-infective towards this strain, a lesser decrease in thickness and biomass probably associated with the bound depolymerases activity is observed, which validates our hypothesis that bacteriophages with high burst size and depolymerase activity have antibiofilm activity regardless of their specificity towards the biofilm-producing host specie.

### 3.7. In Vivo Phage Therapy in G. mellonella Infection Model

For in vivo studies, initially, IBEC40 (non-specific for *E. coli* CFT073) and IBEC 77 (specific) were selected to determine the efficiency of endotoxin removal on the survival of *G. mellonella* larvae at 10^3^ and 10^5^ PFU concentrations. Later, following confirmation of endotoxin removal and toxicity for each phage, they were tested against a high-biofilm-producing *E. coli* CFT073 strain at the different host and phage concentrations. Effective phage and host concentration for prolonged survival was determined compared to the *E. coli* CFT073 strain alone in *G. mellonella* larvae.

It was observed that out of the two phages used for the in vivo experiments, IBEC 40 was not cross-infective towards the *E. coli* CFT073 strain. Similarly, endotoxin removal was successfully achieved to determine the effective survival rate in the *G. mellonella* infection model as per the Moya-Andérico protocol, as shown in Figure 6 [50]. It was observed that the *E. coli* CFT073 strain alone was toxic to *G. mellonella* larvae at a concentration of 4.3 × 10^5^ CFU and above (data not shown). Since IBEC 77 phage and IBEC 40 were not toxic to the *G. mellonella* larvae alone at 1 × 10^3^–10^5^ PFU concentration for over 48 h (Figure 6A), this phage stock was used for further in vivo phage therapy experiments. The effective survival of *G. mellonella* larvae was increased significantly for phage-treated larvae compared to the bacterial host alone, as shown in Figure 6B. For *E. coli* CFT073, the time where 50% of larvae had already died was 22 h. In the case of IBEC77 (×1), this time was increased up until 38 h, while for IBEC77 (×2) it increased to 50.5 h. For IBEC40 (×1) we are talking of 28.5 h, while for IBEC40 (×2) it is reduced in this case to 23 h. Therefore, we reaffirm that the most effective phage is IBEC77, with two doses.

After validating the effect of non-specific phage IBEC 40 by a significant reduction in thickness and biomass in the in-flow biofilm experiment and demonstrating the anti-biofilm activity of naturally available phage-bound depolymerases (acting as natural enzyme-phage bioconjugates), we aimed to evaluate if the efficiency of phage therapy can be improved through the synergistic effect of phage–antibiotic combinations. Therefore, we tested the IBEC 40 and IBEC 77 phages in combination with ciprofloxacin antibiotic against the high-biofilm-producing *E. coli* CFT073 strain in vivo. It was observed that the combination of both ciprofloxacin with the phages (IBEC 40 or IBEC 77) in all tested possibilities (see Section 2.10 and Figure 6B) improves *G. mellonella* survival as 100% of larvae survived for the 55 h that were monitored. However, ciprofloxacin on its own already maintains 100% larva survival. For future studies, new treatment combinations with lower doses of antibiotics and higher phage concentration should be tested.

## 4. Discussion

Considering the importance of alternatives for multi-drug-resistant pathogens and increasing interest in the development of effective in vivo phage therapy approaches, we have reported some of the critical factors limiting phage therapies for in vivo applications. In the last few decades, phage therapy has been significantly explored as an alternative treatment option for antibiotics along with phage lysin-derived engineered peptides [56] as well as phage lytic enzymes [57]. Additionally, due to the newly reported studies, the internalization of phages due to the internalizing peptide available on their surface also restricts efficient applications of in vivo phage therapies [58] and for various clinical applications, including phage therapy in cancer patients [59,60].

Our work highlights the selection and optimization of phages for in vivo phage therapeutic applications. We demonstrated this by isolating effective phages from different sources, including urine, sputum, and sewage, against biofilm-forming selective uropathogenic and invasive biofilm-forming *E. coli* strains, namely *E. coli* CFT073, *E. coli* LF82, and *E. coli* MG1655. We classified the biofilm-forming potential of these selected strains into strong, moderate, and weak as per Stepanović et al. [48], and based on plaque morphology, we selected IBEC 36, IBEC 40, IBEC 48, and IBEC 77 as the candidate phages for further studies. The selection was also based on the specific properties of selected phages, such as the clear and double-layered bull-eyed plaque formation observed for IBEC 40 and IBEC 77. Since this is an indication of depolymerase activity associated with phages, they were considered to be potentially most effective against biofilm-producing pathogens. As biofilms are barriers to bacteriophage therapy and biofilm aging further hinders phage lytic activity as reported previously [34], our work validates that the phages which demonstrate plaque depolymerase activity with high burst size are effective candidates for biofilm disruption as well as infection therapy as observed from our in vitro as well as in vivo experiments with IBEC 40 and IBEC 77 against stronger biofilm-forming host *E. coli* CFT073.

The importance of plaque morphology in the selection of phages for in vivo applications is less well known. The use of phages for in vivo applications, e.g., to treat bacterial infections in animals or humans, requires a detailed understanding of the plaque morphology of the phages used. This is because plaque morphology can provide important information about the virulence and efficacy of phages, as well as their potential for adverse effects, as observed in our in vivo studies with *G. mellonella* larvae.

Clear plaques are generally considered the most desirable for in vivo applications because they are formed by virulent phages that efficiently lyse bacterial cells, which can lead to rapid clearance of infection [61], but the double-layered/bull-eyed morphology of phages is less explored for phage therapies. It should be noted that in vivo applications of phages are still being explored in a controlled manner and there are many unknowns regarding their safety and efficacy as therapeutic agents. It is also important to note that the criteria for selecting phages suitable for in vivo applications are not limited to plaque morphology [62]; other factors such as host range, specificity, stability, and survivability in the host environment must also be considered [61]. 

To avoid false-positive and false-negative results during in vivo studies [63], endotoxins were eliminated from purified and enriched phages. For this, the endotoxin levels were kept far below 0.05 EU/mL for the concentrated phages, which were further diluted with the help of sterilized phosphate buffer (or any suitable buffer and/or water) to eliminate any traces of detectable endotoxins (if present). Since Gram-negative bacterial pathogens are reported to constitute most of the endotoxins, which are even associated with phages following enrichment, endotoxin removal is a mandatory step for successful phage therapies in clinical settings [64]. It is known that even after successful phage therapies against Gram-negative bacterial pathogens, endotoxins are released and are responsible for instigating pro-inflammatory and infusion-related reactions (i.e., hypersensitivity and cytokine release syndromes) and are thus responsible for endotoxic shock. Hence, quantification of endotoxin is very crucial before the implementation of bacteriophage therapy in vitro [61], in animal models as well as in clinical settings to avoid [63,64,65].

Furthermore, bacterial debris and biofilms’ aging can contribute significantly to the endotoxin levels following phage lytic activity, which was responsible for the instantaneous death of *G. mellonella* larvae (within a few minutes) based on the growing stages of *E. coli* host strains. Hence, it is always recommended to quantify endotoxins depending upon the Gram-negative bacterial pathogen responsible for their production, as in the case of *E. coli* strains, which are already reported to produce varying lipopolysaccharide (LPS) complexes in their outer membrane by responding to the factors hindering viability [66,67,68]. Biofilms account for over 80% of human chronic infections and provide resistance against antibiotics and immune response, as they help avoid phagocytosis [1,2]. Since phages and their enzymes are reported to be effective antibiofilm agents [34], phages showing depolymerase activities as reported in this study seem to be a good alternative for biofilm dispersal and antibacterial potential. We believe the effect of phages over the biofilm is similar to the antibiotic treatment by eliminating the bacterial cells located on the biofilm surfaces to produce clear zones in the extracellular biofilm matrix that increase their cell removal in the overall structure.

Both IBEC 40 and IBEC 77 bacteriophages have a positive effect in the treatment of infected *G. mellonella* larvae with the uropathogenic strain *E. coli* CFT073 as they significantly increase the survival of larvae in comparison with the larvae only infected with the bacteria. An interesting fact was the complete and quick melanization of all infected larvae, mainly due to the high bacterial dose of infection. Thus, infected larvae were also more compromised as more doses (which means more punctures) were received. For this, two doses did not always give better results than one single dose. Previously, we had already tested these therapies with three doses (data not shown), but the same phenomena occurred.

Although these phages alone cannot completely avoid larvae mortality, excellent results were obtained with the different combinations of the phages with ciprofloxacin, as it resulted in the total survival of larvae. Being in the spotlight, phage therapy is, therefore, a promising approach. However, further in vivo studies are necessary to understand the synergistic effects of phage-antibiotics treatment therapies to know their effects on biofilm-producing Gram-negative bacterial pathogens for biofilm dispersal and eradication of infections.

## 5. Conclusions

Clinically isolated bacteriophages IBEC40 and IBEC77 demonstrated efficient removal of biofilm produced by *E. coli* CFT073 strain during in vitro in-flow biofilm, and they also showed prolonged survival in *G. mellonella* larvae after 55 h. Our work demonstrates that bacteriophages with high burst size and depolymerase activity can be a good alternative for treating uropathogenic and invasive biofilm-forming *E. coli* pathogens. Further studies are necessary to understand if the efficiency of phage therapy in vivo in biofilm-producing bacterial pathogens can be improved synergistically in combination with antibiotics. Additionally, in line with our hypothesis, phage-bound enzymes must be explored for their antibiofilm as well as antibacterial activity irrespective of their specificity towards their hosts.

## Figures and Tables

**Figure 1 cells-12-00344-f001:**
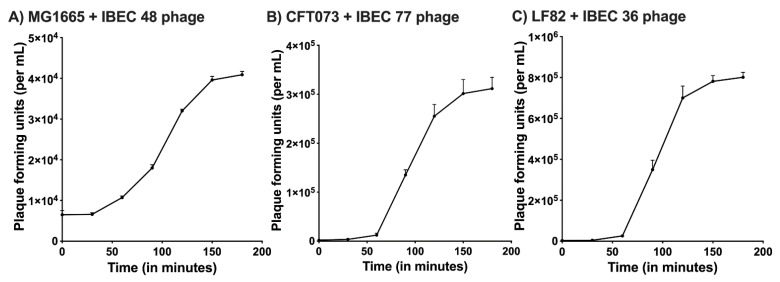
One-step growth curve of (**A**) IBEC48 phage, (**B**) IBEC77 phage, and (**C**) IBEC36 phage. One-step growth curves were obtained by growing IBEC 48, IBEC 77, and IBEC 36 with there exponentially growing culture of *E. coli* MG1655, *E. coli* CFT073, and *E. coli* LF82 strains respectively. Data points indicate the PFU/mL at different time points. Each data point represents the mean of three independent experiments.

**Figure 2 cells-12-00344-f002:**
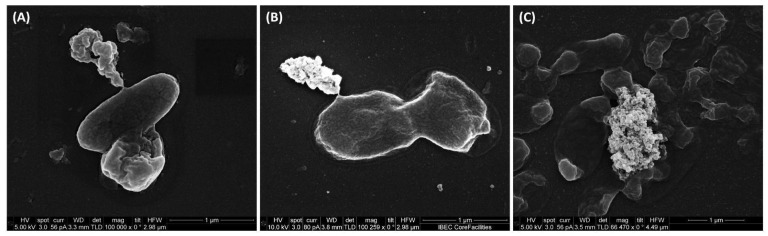
Representative scanning electron microscopy images of (**A**) *E. coli* CFT073, (**B**) *E. coli* LF82, and (**C**) *E. coli* MG1655 lysed by IBEC 77, IBEC 36, and IBEC 48 phage, respectively, and releasing its intracellular content as observed in these images. Bar = 1 µm.

**Figure 3 cells-12-00344-f003:**
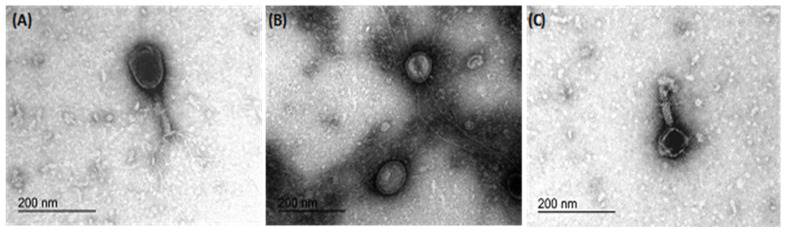
Representative transmission electron microscopy micrographs of bacteriophages (**A**) IBEC 36, (**B**) IBEC 77, and (**C**) IBEC 48 isolated from clinical samples against *E. coli* LF82, *E. coli* CFT073, and *E. coli* MG1655, respectively. All were only active against the representative strains and were not cross-reactive. (Bar = 200 nm).

**Figure 4 cells-12-00344-f004:**
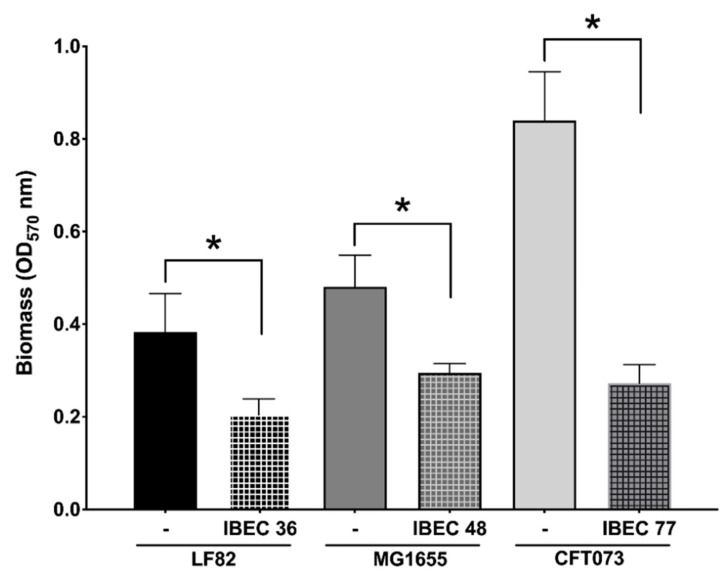
Effect of relative in vitro biofilm reduction in *E. coli* CFT073, *E. coli* LF82, and *E. coli* MG1655 by IBEC77, IBEC36, and IBEC48 phages, respectively, with phage concentration of 1 × 10^6^ pfu/mL after 3-day-old biofilm (72 h) growth in 96-well plates at 37 °C. Standard deviations are indicated. (*) Statistically significant differences representing (* *p* < 0.0001) were determined by unpaired *t*-test (GraphPad Prism v.9).

**Figure 5 cells-12-00344-f005:**
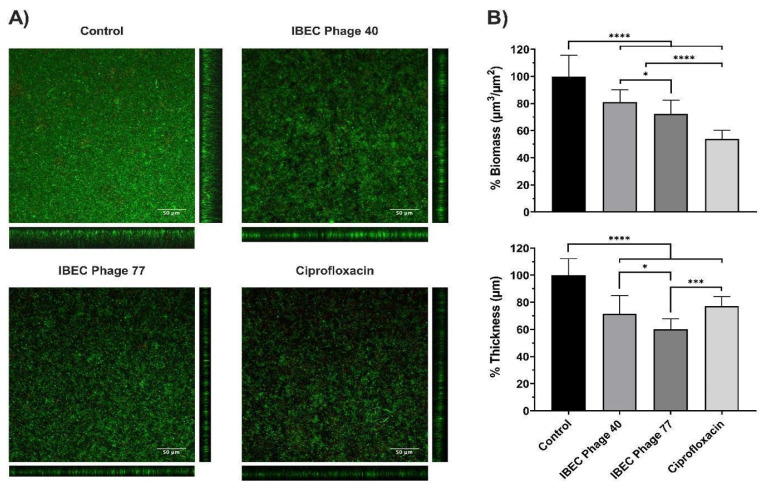
The relative effect of *E. coli* CFT073 specific IBEC77 and *E. coli* CFT073 non-specific IBEC 40 against 96 h biofilms formed under flow conditions. Phages at 1 × 10^7^ pfu/mL, Ciprofloxacin 1 µg/mL, and LB Broth were employed as treatments. (**A**) Confocal microscopy images (sum and orthogonal views) of stained biofilms and (**B**) biomass and average thickness reduction with respect to control. Standard deviations of data from two different replicates are indicated. Statistically significant differences (* *p* < 0.05, *** *p* < 0.001, **** *p* < 0.0001) were determined by an ordinary one-way ANOVA (GraphPad Prism v.9).

**Figure 6 cells-12-00344-f006:**
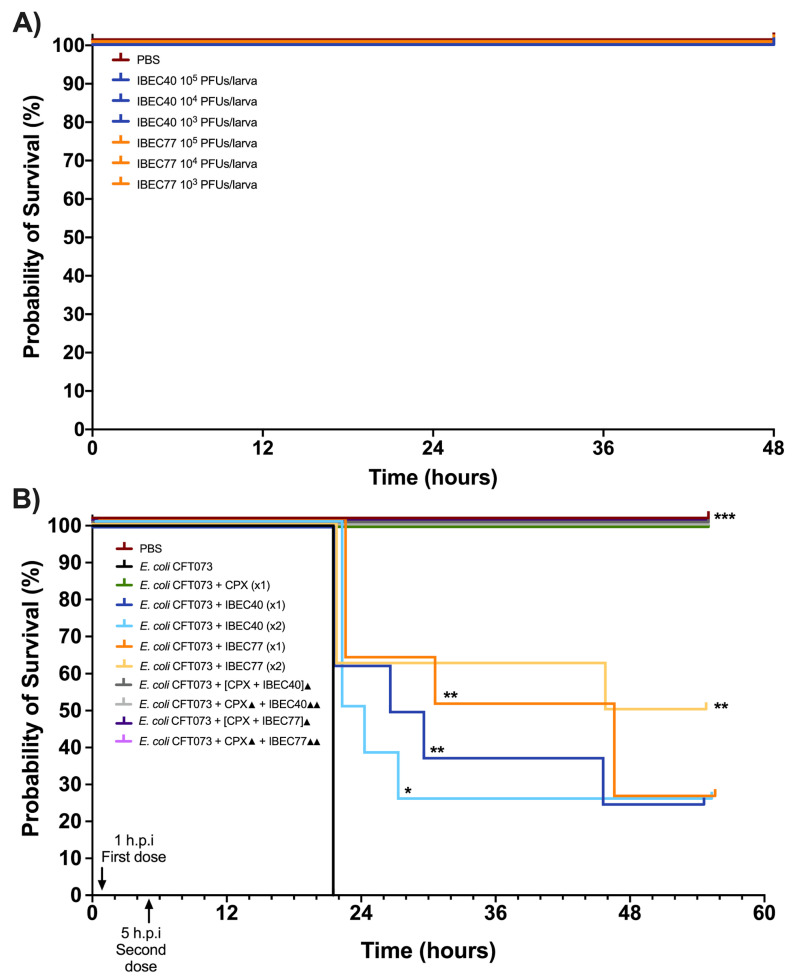
In vivo toxicity of *E. coli* CFT073, IBEC 40, and IBEC 77 in *G. mellonella* larvae and effective survival concentration for in vivo phage therapy. (**A**) Different concentrations of IBEC 40 and IBEC 77 phages showed no toxic effects on the survival of *G. mellonella* larvae over 48 h of inoculation. (**B**) Survival curve of *G. mellonella* larvae with effective concentrations of *E. coli* CFT073 and IBEC 40 and IBEC 77 phages alone and in different combinations with ciprofloxacin (20 mg/kg larva) for studying the prolongation of larvae survival, compared with the bacterial host strain alone. For the individual treatments of phages, the clarification x1 indicates one dose at 1 h post-infection (h.p.i) and the clarification ×2 indicates two doses (the first dose at 1 h.p.i and the second dose at 5 h.p.i). In the combined treatments of phages with the antibiotic, the ▲ clarification in the legend indicates one dose at 1 h.p.i and ▲▲ indicates 2 h.p.i. Larvae were monitored up until 55 h after inoculation. Graphs were plotted with GraphPad Prism version 9.0. Statistically significant differences (* *p* < 0.05, ** *p* < 0.01, *** *p* < 0.001) between all treatments and the *E. coli* CFT073 were determined with GraphPad Prism version 9.0, by using the Long-rank test and Gehan–Breslow–Wilcoxon test.

**Table 1 cells-12-00344-t001:** Different phages were isolated from clinical and environmental samples.

Phage Number	Host Strain	Source	Specific Property	Turbidity	Plaque Diameter	Geographical Location
IBEC 36	*E. coli* LF82	Urine	NO	Clear	Large (>1 cm)	Barcelona (Spain)
IBEC 37	*E. coli* LF82	Urine	NO	Blur	Large (>1 cm)	Barcelona (Spain)
IBEC 38	*E. coli* LF82	Urine	NO	Blur	Large (>1 cm)	Barcelona (Spain)
IBEC 39	*E. coli* LF82	Sputum	NO	Clear	Large (>1 cm)	Barcelona (Spain)
IBEC 40	*E. coli* LF82	Sputum	Double-layered, Bull-eyed	Clear	Small (<0.5 cm)	Barcelona (Spain)
IBEC 41	*E. coli* MG1655	Sewage	NO	Clear	Small (<0.5 cm)	Barcelona (Spain)
IBEC 42	*E. coli* MG1655	Sewage	NO	Clear	Small (<0.5 cm)	Barcelona (Spain)
IBEC 43	*E. coli* MG1655	Sewage	NO	Clear	Small (<0.5 cm)	Barcelona (Spain)
IBEC 44	*E. coli* MG1655	Sewage	NO	Clear	Large (>1 cm)	Barcelona (Spain)
IBEC 45	*E. coli* MG1655	Sewage	NO	Clear	Small (<0.5 cm)	Barcelona (Spain)
IBEC 46	*E. coli* MG1655	Sputum	NO	Blur	Small (<0.5 cm)	Barcelona (Spain)
IBEC 47	*E. coli* MG1655	Sputum	NO	Blur	Small (<0.5 cm)	Barcelona (Spain)
IBEC 48	*E. coli* MG1655	Urine	Double-layered, Bull-eyed	Clear	Large (>1 cm)	Barcelona (Spain)
IBEC 49	*E. coli* MG1655	Urine	NO	Clear	Medium (0.5–1 cm)	Barcelona (Spain)
IBEC 50	*E. coli* MG1655	Urine	NO	Clear	Small (<0.5 cm)	Barcelona (Spain)
IBEC 51	*E. coli* MG1655	Sputum	NO	Clear	Large (>1 cm)	Barcelona (Spain)
IBEC 52	*E. coli* MG1655	Sputum	NO	Clear	Medium (0.5–1 cm)	Barcelona (Spain)
IBEC 53	*E. coli* MG1655	Sputum	NO	Clear	Small (<0.5 cm)	Barcelona (Spain)
IBEC 77	*E. coli* CFT073	Urine	Double-layered, Bull-eyed	Clear	Medium (0.5–1 cm)	Barcelona (Spain)
IBEC 78	*E. coli* CFT073	Sewage	NO	Clear	Medium (0.5–1 cm)	Barcelona (Spain)
IBEC 98	*E. coli* CFT073	Urine	Double-layered, Bull-eyed	Clear	Medium (0.5–1 cm)	Barcelona (Spain)

**Note:** Average plaque diameter was determined after overnight incubation at 37 °C and were classified into small, medium, and large plaques for differentiating plaques by plaque morphology.

**Table 2 cells-12-00344-t002:** Cross-infective phages against different *E. coli* clinically isolated strains. Original host strains (clear zone of lysis, OH: blue), susceptible bacterial strains (clear zone of lysis, S: green), and resistant strains (no zone of lysis, R: red) against phage infection.

Bacteriophage Number	Original Host	*E. coli* LF82	*E. coli* MG1655	*E. coli* CFT073
IBEC 36	*E. coli* LF82	**OH**	**R**	**R**
IBEC 37	*E. coli* LF82	**OH**	**R**	**R**
IBEC 38	*E. coli* LF82	**OH**	**R**	**R**
IBEC 39	*E. coli* LF82	**OH**	**R**	**R**
IBEC 40	*E. coli* LF82	**OH**	**S**	**R**
IBEC 41	*E. coli* MG1655	**R**	**OH**	**R**
IBEC 42	*E. coli* MG1655	**R**	**OH**	**R**
IBEC 43	*E. coli* MG1655	**R**	**OH**	**R**
IBEC 44	*E. coli* MG1655	**R**	**OH**	**R**
IBEC 45	*E. coli* MG1655	**R**	**OH**	**R**
IBEC 46	*E. coli* MG1655	**R**	**OH**	**R**
IBEC 47	*E. coli* MG1655	**R**	**OH**	**R**
IBEC 48	*E. coli* MG1655	**R**	**OH**	**R**
IBEC 49	*E. coli* MG1655	**R**	**OH**	**R**
IBEC 50	*E. coli* MG1655	**R**	**OH**	**R**
IBEC 51	*E. coli* MG1655	**R**	**OH**	**R**
IBEC 52	*E. coli* MG1655	**R**	**OH**	**R**
IBEC 53	*E. coli* MG1655	**R**	**OH**	**R**
IBEC 77	*E. coli* CFT073	**R**	**R**	**OH**
IBEC 78	*E. coli* CFT073	**R**	**R**	**OH**
IBEC 98	*E. coli* CFT073	**R**	**R**	**OH**

**Note:** For the cross infection studies, the average repeates performed for each *E. coli* strain against each isolated IBEC phages were minimum of 3 times. Here, the phages which formed blur or turbid plaques were not considered as cross infective for this studies for selection of the best candidate phages for further studies. The spot test for cross-infectivity study were determined after overnight incubation at 37 °C.

## Data Availability

Data is contained within the article or Appendix A. The data presented in this study are available in the article or Appendix A.

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
