# Peer review of "Accessing the In Vivo Efficiency of Clinically Isolated Phages against Uropathogenic and Invasive Biofilm-Forming Escherichia coli Strains for Phage Therapy"

_cells, 2023, doi:10.3390/cells12030344_

Round 1
Reviewer 1 Report
The manuscript titled “Accessing the in vivo efficiency of clinically isolated phages against uropathogenic and invasive biofilm-forming Escherichia coli strains for phage therapy” by Sanmukh et al. describes the isolation of phages clinical settings against clinical strains of E.coli. It is an interesting study of high clinical relevance. However, I have certain concerns.
I receommend major revision before the manuscript is accepted for publication
My Comments are as follows:
1. Abstract:
“irrespective of host specificity, bacteriophages producing clear plaques with a short latency period, high burst size, and exhibiting depolymerizes activity, are the best candidates against biofilm-producing E. coli pathogens....”
-Without host specificity a phage cannot infect a host, thus cannot form plaque. IT would great if authors can clarify exactly what they want to convey.
2. Introduction:
Line 35-36: “In chronic infection treatments, biofilms pose a significant problem due to their reduced antibiotic sensitivity, and therefore approximately 80% of bacterial infections are caused by biofilms”
-Infection persistence and recurrence of infections are associated with reduced drug sensitivity of biofilm residing bacteria but can biofilm be framed as primary causative factor of bacterial infections in human because of reduced antibiotic sensitivity?
This sentence very misleading and authors must rephrase it.
Line 40: “Biofilm resistance is one of the major modes of drug resistance transmission in most biofilm-forming bacteria, including E. coli”
-Biofilm mediated drug/antibiotic resistance is more appropriate than Biofilm resistance....
Line 49-50: Diffusely adherent E. coli, uropathogenic E. coli, Meningitis-associated E. coli
-E. coli should be italicized throughout the text
Line55: “...therapies in BALB/c mice and Galleria mellonella larvae...”
- Galleria mellonella should be italicized
Line 79: matrix produced by some microbial species, i.d. Staphylococcus aureus, E. coli, Klebsiella pneumoniae, Pseudomonas aeruginosa
-i.e., in place of i.d.
Line: In the introduction; only disadvantages of phage therapy are mentioned, no supporting literature on phage therapy against biofilm is stated or cited.
3. Material & Methods
Section 2.2 Processing of clinical samples for phage isolation:
Source of Clinical samples for phage isolation is not mentioned clearly
Section 2.3 Phage propagation and lysate preparation:
a. “Optical density reached 0.5 (OD550 = ± 0.5)”
-How can the error margin be +/- 0.5 if the O.D itself is 0.5?
b. “the supernatant obtained above was inoculated at 30°C with shaking until complete lysis occurred”
-At what MOI/quantity of supernatant containing phage particles was added to the 500 ml host culture
for the production of lab scale phage particle?
c. “the supernatant obtained above was inoculated at 30°C with shaking until complete lysis occurred”
- How does complete lysis was determined/evaluated has not been mentioned. Is it corresponding to certain O.D / Viable bacterial cell count, nothing has been clarified.
Section 2.4 Endotoxin removal and Bacteriophages purification:
-Reference article cited [42] for two-step differential PEG precipitation doesn’t contain the method.
-The methodology is not clearly described.
Section 2.6 Phage one-step growth characteristics:
Line 162: “The different multiplicity of infection (MOI) within the range of 0.1 to 0.0001 were selected to determine the effective phage-host concentration”
Why different MOIs were used to evaluate the one step growth curve of the phages? Anyway, it will not affect the latency or the burst size of the isolated phages.
Section 2.7 Scanning Electron Microscopy (SEM) for studying the lytic phage activity:
“500 ml of each E. coli strain grown overnight with OD600= 0.1 were inoculated with their respective phages with a minimum concentration of 106 pfu/ml were incubated overnight at 37 °C constant shakings at 200 rpm in a 50 ml falcon flask.”
-The statement is not clear. Please clarify
-What was the MOI / quantity of bacteriophage used in these experiments?
4. Results
Section 3.1 Isolation and Selection of phages:
“3 selection criteria; clear plaque morphology, double halo formation (bull eye morphology), and/or both”
-Phage selection criteria are not clear as all plaques with bull eye morphology are clear plaques.
- In supplementary information only Plaque image of IBEC77 is provided, plaque morphology of other two selected phages should be included.
- As mentioned in the manuscript, plaque of IBEC77 is of bull eye morphology (because of the phage depolymerase activity). However, in SI the plaque image of IBEC77 is not showing any depolymerase activity and bull eye formation. Probably the image was taken in a short period of time, post spotting and depolymerase activity was not enhanced enough to be visible. However, the plaque formation 24
hours post spotting could have contain emerged phage resistant small colonies, which could have contradicted the criteria of clear plaque morphology.
Section 3.3 Phage one-step growth curve:
“For IBEC 36 and IBEC 77, a multiplicity of infection (MOI) used to determine their one-step growth curve was 0.0001, whereas, for IBEC 48, the MOI was 0.001 due to its long latency period and small burst size”
- Before performing the one step growth curve how the latency period and burst size was determined? and the optimum MOI was set for performing one step growth curve? Does MOI anyway modulate
latency period and small burst size?
- Further platting should be done till complete stationary phase is achieved. As the phages were still bursting out and plateau phase was not achieved
Section 3.4 Scanning Electron Microscopy studies:
-Control (phage uninfected cell)s should be kept. Please provide the images
Section 3.6: In-vitro Biofilm assay:
-Biofilm contains bacterial cells and cell secreted biofilm matrix. Biomass degradation does not necessarily indicate that it is reducing the viability. In case of nonspecific phage, IBEC4O treatment; the phage secreted depolymerase is possibly reducing the biofilm matrix but not lysing the bacterial cells. To check the impact of phages on cellular level of biofilm, CFU should be performed Post treatment. Please, provide the viability data
Section 3.7 In-vivo phage therapy in G. mellonella infection model:
“It was observed that out of the 2 phages used for the in vivo experiments, IBEC40 was cross-infective, whereas IBEC77 was not cross-infective with the E. coli CFT073 strain.”
- Earlier in the manuscript it is mentioned that IBEC40 was cross infective against MG1655 and not specific against CFT073 whereas IBEC77 was specific to CFT073. Please clarify.
- What concentration of ciprofloxacin was used?
- 1 dose of ciprofloxacin alone is preventing the larval death throughout the observation period indicating an efficient treatment intervention. However single or double dose of both of the specific or non-specific phages reduced survival of the larva. As ciprofloxacin alone itself is efficiently inhibiting larval death, combining the ciprofloxacin with any of the phages does not evaluate phage efficacy, hence the (antibiotic + phage) combination study is not relevant since the phage toxicity test is already been evaluated.
-1X indicates one dose at 1 h post-infection (h.p.i) â–² also indicates one dose at 1 h post-infection (h.p.i).
5. Conclusions
“Our work demonstrates that bacteriophages with a short latency period and high burst size, with depolymerized activity, can be the best alternative for treating uropathogenic and invasive biofilm-forming E. coli pathogens.”
- As any comparative study (phage with long latency, small burst size and without depolymerase activity) was not perform then how does this study can be concluded with afore mentioned statement.
Suggestions:
- Bacterial Challenge test with different MOI must be included to evaluate the infectivity curve of the phages against bacteria.
- In-vitro static antibiofilm assay with specific /nonspecific phages and phage antibiotic combination to correctly evaluate the anti-biofilm efficacy need to done.
The manuscript titled “Accessing the in vivo efficiency of clinically isolated phages against
uropathogenic and invasive biofilm-forming Escherichia coli strains for phage therapy” by
Sanmukh et al. describes the isolation of phages clinical settings against clinical strains of E.coli.
It is an interesting study of high clinical relevance. However, I have certain concerns.
I receommend major revision before the manuscript is accepted for publication
My Comments are as follows:
1. Abstract:
“irrespective of host specificity, bacteriophages producing clear plaques with a short latency period,
high burst size, and exhibiting depolymerizes activity, are the best candidates against biofilm-producing
E. coli pathogens....”
-Without host specificity a phage cannot infect a host, thus cannot form plaque. IT would great if authors
can clarify exactly what they want to convey.
2. Introduction:
Line 35-36: “In chronic infection treatments, biofilms pose a significant problem due to their reduced
antibiotic sensitivity, and therefore approximately 80% of bacterial infections are caused by biofilms”
-Infection persistence and recurrence of infections are associated with reduced drug sensitivity of
biofilm residing bacteria but can biofilm be framed as primary causative factor of bacterial infections
in human because of reduced antibiotic sensitivity?
This sentence very misleading and authors must rephrase it.
Line 40: “Biofilm resistance is one of the major modes of drug resistance transmission in most biofilm-
forming bacteria, including E. coli”
-Biofilm mediated drug/antibiotic resistance is more appropriate than Biofilm resistance....
Line 49-50: Diffusely adherent E. coli, uropathogenic E. coli, Meningitis-associated E. coli
-E. coli should be italicized throughout the text
Line55: “...therapies in BALB/c mice and Galleria mellonella larvae...”
- Galleria mellonella should be italicized
Line 79: matrix produced by some microbial species, i.d. Staphylococcus aureus, E. coli, Klebsiella
pneumoniae, Pseudomonas aeruginosa
-i.e., in place of i.d.
Line: In the introduction; only disadvantages of phage therapy are mentioned, no supporting literature
on phage therapy against biofilm is stated or cited.
3. Material & Methods
Section 2.2 Processing of clinical samples for phage isolation:
Source of Clinical samples for phage isolation is not mentioned clearly
Section 2.3 Phage propagation and lysate preparation:
a. “Optical density reached 0.5 (OD550 = ± 0.5)”
-How can the error margin be +/- 0.5 if the O.D itself is 0.5?
b. “the supernatant obtained above was inoculated at 30°C with shaking until complete lysis occurred”
-At what MOI/quantity of supernatant containing phage particles was added to the 500 ml host culture
for the production of lab scale phage particle?
c. “the supernatant obtained above was inoculated at 30°C with shaking until complete lysis occurred”
- How does complete lysis was determined/evaluated has not been mentioned. Is it corresponding to
certain O.D / Viable bacterial cell count, nothing has been clarified.
Section 2.4 Endotoxin removal and Bacteriophages purification:
-Reference article cited [42] for two-step differential PEG precipitation doesn’t contain the method.
-The methodology is not clearly described.
Section 2.6 Phage one-step growth characteristics:
Line 162: “The different multiplicity of infection (MOI) within the range of 0.1 to 0.0001 were selected
to determine the effective phage-host concentration”
Why different MOIs were used to evaluate the one step growth curve of the phages? Anyway, it will
not affect the latency or the burst size of the isolated phages.
Section 2.7 Scanning Electron Microscopy (SEM) for studying the lytic phage activity:
“500 ml of each E. coli strain grown overnight with OD600= 0.1 were inoculated with their respective
phages with a minimum concentration of 106 pfu/ml were incubated overnight at 37 °C constant
shakings at 200 rpm in a 50 ml falcon flask.”
-The statement is not clear. Please clarify
-What was the MOI / quantity of bacteriophage used in these experiments?
4. Results
Section 3.1 Isolation and Selection of phages:
“3 selection criteria; clear plaque morphology, double halo formation (bull eye morphology), and/or
both”
-Phage selection criteria are not clear as all plaques with bull eye morphology are clear plaques.
- In supplementary information only Plaque image of IBEC77 is provided, plaque morphology of other
two selected phages should be included.
- As mentioned in the manuscript, plaque of IBEC77 is of bull eye morphology (because of the phage
depolymerase activity). However, in SI the plaque image of IBEC77 is not showing any depolymerase
activity and bull eye formation. Probably the image was taken in a short period of time, post spotting
and depolymerase activity was not enhanced enough to be visible. However, the plaque formation 24
hours post spotting could have contain emerged phage resistant small colonies, which could have
contradicted the criteria of clear plaque morphology.
Section 3.3 Phage one-step growth curve:
“For IBEC 36 and IBEC 77, a multiplicity of infection (MOI) used to determine their one-step growth
curve was 0.0001, whereas, for IBEC 48, the MOI was 0.001 due to its long latency period and small
burst size”
- Before performing the one step growth curve how the latency period and burst size was determined?
and the optimum MOI was set for performing one step growth curve? Does MOI anyway modulate
latency period and small burst size?
- Further platting should be done till complete stationary phase is achieved. As the phages were still
bursting out and plateau phase was not achieved
Section 3.4 Scanning Electron Microscopy studies:
-Control (phage uninfected cell)s should be kept. Please provide the images
Section 3.6: In-vitro Biofilm assay:
-Biofilm contains bacterial cells and cell secreted biofilm matrix. Biomass degradation does not
necessarily indicate that it is reducing the viability. In case of nonspecific phage, IBEC4O treatment;
the phage secreted depolymerase is possibly reducing the biofilm matrix but not lysing the bacterial
cells. To check the impact of phages on cellular level of biofilm, CFU should be performed Post
treatment. Please, provide the viability data
Section 3.7 In-vivo phage therapy in G. mellonella infection model:
“It was observed that out of the 2 phages used for the in vivo experiments, IBEC40 was cross-infective,
whereas IBEC77 was not cross-infective with the E. coli CFT073 strain.”
- Earlier in the manuscript it is mentioned that IBEC40 was cross infective against MG1655 and not
specific against CFT073 whereas IBEC77 was specific to CFT073. Please clarify.
- What concentration of ciprofloxacin was used?
- 1 dose of ciprofloxacin alone is preventing the larval death throughout the observation period
indicating an efficient treatment intervention. However single or double dose of both of the specific or
non-specific phages reduced survival of the larva. As ciprofloxacin alone itself is efficiently inhibiting
larval death, combining the ciprofloxacin with any of the phages does not evaluate phage efficacy,
hence the (antibiotic + phage) combination study is not relevant since the phage toxicity test is already
been evaluated.
-1X indicates one dose at 1 h post-infection (h.p.i) â–² also indicates one dose at 1 h post-infection (h.p.i).
5. Conclusions
“Our work demonstrates that bacteriophages with a short latency period and high burst size, with
depolymerized activity, can be the best alternative for treating uropathogenic and invasive biofilm-
forming E. coli pathogens.”
- As any comparative study (phage with long latency, small burst size and without depolymerase
activity) was not perform then how does this study can be concluded with afore mentioned statement.
Suggestions:
- Bacterial Challenge test with different MOI must be included to evaluate the infectivity curve of the
phages against bacteria.
- In-vitro static antibiofilm assay with specific /nonspecific phages and phage antibiotic combination to
correctly evaluate the anti-biofilm efficacy need to done.
Author Response
See enclosed file
Dear Reviewer 1
Thank you for thoroughly reviewing our paper and for the opportunity to submit a revised version. We much appreciate the reviewer's constructive comments on our manuscript (Manuscript cells-2083777, Accessing the in vivo efficiency of clinically isolated phages against uropathogenic and invasive biofilm-forming Escherichia coli strains for phage therapy), which have been of great help and have improved the manuscript over the previous version. We are really thankful for your work on our manuscript.
Our responses to his/her comments are detailed below (in red).
With the manuscript changes detailed below and our answers to the reviewer's comments, we hope you will now find the revised version of our manuscript acceptable for publication in Cells.
Sincerely,
Dr. Eduard Torrents
Response to Reviewer 1:
The manuscript titled “Accessing the in vivo efficiency of clinically isolated phages against uropathogenic and invasive biofilm-forming Escherichia coli strains for phage therapy” by Sanmukh et al. describes the isolation of phages clinical settings against clinical strains of E.coli. It is an interesting study of high clinical relevance. However, I have certain concerns.
I receommend major revision before the manuscript is accepted for publication
My Comments are as follows:
- Abstract:
“irrespective of host specificity, bacteriophages producing clear plaques with a short latency period, high burst size, and exhibiting depolymerizes activity, are the best candidates against biofilm-producing E. coli pathogens....”
-Without host specificity a phage cannot infect a host, and thus cannot form plaque. IT would great if authors can clarify exactly what they want to convey.
Response: Thank you for your comments. We used specific and non-specific phages for our reported in-vitro and in-vivo studies. In both cases, phages showing depolymerase activities were used and were effective in reducing the biofilm load and enhancing survival in our in-vivo model. Therefore, we suggest that phage-bound depolymerase activity is effective irrespective of host specificity, even if these phages are not specific against their host. It has nothing to do with host infection or plaque formation. We have clarified it again in the abstract as well as the result and discussion as per your suggestions.
- Introduction:
Line 35-36: “In chronic infection treatments, biofilms pose a significant problem due to their reduced antibiotic sensitivity, and therefore approximately 80% of bacterial infections are caused by biofilms”
-Infection persistence and recurrence of infections are associated with reduced drug sensitivity of biofilm residing bacteria but can biofilm be framed as primary causative factor of bacterial infections in human because of reduced antibiotic sensitivity?
This sentence very misleading and authors must rephrase it.
Response: Thank you for your critical suggestion. We agree with you, and we have modified these sentences. See the new sentences in the introduction section.
Line 40: “Biofilm resistance is one of the major modes of drug resistance transmission in most biofilm-forming bacteria, including E. coli”
-Biofilm mediated drug/antibiotic resistance is more appropriate than Biofilm resistance....
Response: Thank you for your critical suggestion. We have modified the statement accordingly.
Line 49-50: Diffusely adherent E. coli, uropathogenic E. coli, Meningitis-associated E. coli
-E. coli should be italicized throughout the text
Response: Thank you for this observation. We have modified the statement accordingly.
Line55: “...therapies in BALB/c mice and Galleria mellonella larvae...”
- Galleria mellonella should be italicized
Response: Thank you, now it is all corrected.
Line 79: matrix produced by some microbial species, i.d. Staphylococcus aureus, E. coli, Klebsiella pneumoniae, Pseudomonas aeruginosa
-i.e., in place of i.d.
Response: Thank you, all corrected.
Line: In the introduction; only disadvantages of phage therapy are mentioned, no supporting literature on phage therapy against biofilm is stated or cited.
Response: Thank you for this observation. We have included new references supporting phage therapy as a potential alternative against biofilm formation/removal.
- Material & Methods
Section 2.2 Processing of clinical samples for phage isolation:Source of Clinical samples for phage isolation is not mentioned clearly
Response: Thank you for your suggestion. We have now included the source of bacteriophage isolation and the ethically approved protocol (PR(AG)275/2019) to transfer clinical samples of waste material to the IBEC laboratories.
Section 2.3 Phage propagation and lysate preparation:
- “Optical density reached 0.5 (OD550 = ± 0.5)”
-How can the error margin be +/- 0.5 if the O.D itself is 0.5?
Response: Thank you, you are right, and we have modified the statement.
- “the supernatant obtained above was inoculated at 30°C with shaking until complete lysis occurred”
-At what MOI/quantity of supernatant containing phage particles was added to the 500 ml host culturefor the production of lab scale phage particle?
Response: Thank you for your critical suggestion. We modified the statement and have included the concentration of phages inoculated at this stage.
- “the supernatant obtained above was inoculated at 30°C with shaking until complete lysis occurred”
- How does complete lysis was determined/evaluated has not been mentioned. Is it corresponding to a certain O.D / Viable bacterial cell count, nothing has been clarified.
Response: Thank you for your critical suggestion. We modified the statement and have included the statement by adding “….until considerable lysis was observed”.
Section 2.4 Endotoxin removal and Bacteriophages purification:
-Reference article cited [42] for two-step differential PEG precipitation doesn’t contain the method.
-The methodology is not clearly described.
Response: Thank you for your critical suggestion. We have now included the correct references and have explained the method in a better way.
Section 2.6 Phage one-step growth characteristics:
Line 162: “The different multiplicity of infection (MOI) within the range of 0.1 to 0.0001 were selected to determine the effective phage-host concentration”
Why different MOIs were used to evaluate the one-step growth curve of the phages? Anyway, it will not affect the isolated phages' latency or burst size.
Response: Thank you, this is a crucial question that helped us to optimize the phage one-step growth curve. We noticed that for some phages, as was observed during our study, varying the MOIs reduced the effects of performing one-step growth characteristics. Since phages have varying burst sizes ranging from 15-20 to many hundreds, we showed the results where we got the best results. As varying MOIs do not affect the latency or the burst size, as you rightly mentioned, we assume that such minor changes can simplify the workload associated with evaluating the one-step growth curve of the phage.
Section 2.7 Scanning Electron Microscopy (SEM) for studying the lytic phage activity:
“500 ml of each E. coli strain grown overnight with OD600= 0.1 were inoculated with their respective phages with a minimum concentration of 106 pfu/ml were incubated overnight at 37 °C constant shakings at 200 rpm in a 50 ml falcon flask.”
-The statement is not clear. Please clarify
-What was the MOI / quantity of bacteriophage used in these experiments?
Response: Thank you for this suggestion. We have provided the concentration of hosts and phages for clarification.
- Results
Section 3.1 Isolation and Selection of phages:
“3 selection criteria; clear plaque morphology, double halo formation (bull eye morphology), and/or both”
-Phage selection criteria are not clear as all plaques with bull eye morphology are clear plaques.
- In supplementary information only Plaque image of IBEC77 is provided, plaque morphology of other two selected phages should be included.
- As mentioned in the manuscript, plaque of IBEC77 is of bull eye morphology (because of the phage depolymerase activity). However, in SI the plaque image of IBEC77 is not showing any depolymerase activity and bull eye formation. Probably the image was taken in a short period of time, post spotting and depolymerase activity was not enhanced enough to be visible. However, the plaque formation 24hours post-spotting could have contain emerged phage-resistant small colonies, which could have contradicted the criteria of clear plaque morphology.
Response: Thank you for this critical observation. We have made the changes to the table, as it was a typo for the phage selection criteria. Since our work is focused on the high biofilm-producing E. coli CFT073 strain, we focus on the phages which affect it and not the others as they were not considered for most in-vitro and in-vivo studies. We used IBEC 36, IBEC 48, and IBEC 77 initial studies associated with phage one-step growth curve, static biofilm removal potential, SEM, and TEM analysis, which are reported, but later, we focused on IBEC 40 and IBEC 77. As IBEC 40 is not specific to the E. coli CFT073 strain but has depolymerase activity, which can be seen in supplementary figure 1, we used it as a non-specific phage with depolymerase activity against E. coli CFT073 strain for in-flow biofilm removal studies as well as in-vivo studies with Galleria mellonella. As IBEC 77 was specific to the E. coli CFT073 strain, we show its plaque morphology along with the effect of IBEC 40 on this strain. As IBEC 36 and IBEC 48 were not potential candidates for these studies, we avoided providing more data related to them.
Section 3.3 Phage one-step growth curve:
“For IBEC 36 and IBEC 77, a multiplicity of infection (MOI) used to determine their one-step growth curve was 0.0001, whereas, for IBEC 48, the MOI was 0.001 due to its long latency period and small burst size”
- Before performing the one step growth curve how the latency period and burst size was determined? and the optimum MOI was set for performing one step growth curve? Does MOI anyway modulate
latency period and small burst size?
- Further platting should be done till complete stationary phase is achieved. As the phages were still bursting out and plateau phase was not achieved
Response: Thank you for this suggestion. We agree with the reviewer regarding the effect of MOI does not modulate the latency period or burst size. Still, we used varying MOIs to facilitate fast detection of the one-step growth curve as well as the latent period. The burst size and latent period were not previously known, but this study modification using different MOIs helps us predict the best possible results. Also, we performed the plating until we got stationary phage; hence, we did not perform further plating as suggested by the reviewer. But we have included the new value based on our newly completed experiments.
Section 3.4 Scanning Electron Microscopy studies:
-Control (phage uninfected cell)s should be kept. Please provide the images
Response: Thank you for this suggestion. The control is provided in Supplementary Figure 3. The control for scanning electron microscopy for phage uninfected E. coli LF82, E. coli MG1655, and E. coli CFT073 biofilm-producing strains.
Section 3.6: In-vitro Biofilm assay:
-Biofilm contains bacterial cells and cell-secreted biofilm matrix. Biomass degradation does not necessarily indicate that it is reducing viability. In case of non-specific phage, IBEC4O treatment; the phage secreted depolymerase is possibly reducing the biofilm matrix but not lysing the bacterial cells. To check the impact of phages on cellular level of biofilm, CFU should be performed Post-treatment. Please, provide the viability data
Response: We agree with the reviewer and do not mention that not specific phages with depolymerase activity reduce the bacterial load by cell lysis. The in-vitro studies include 2 types of studies, 1) Static biofilm and 2) In-flow biofilm experiments. In the static biofilm study, we show the effect of individual phages against 3 bacterial strains. After selecting the best biofilm-producing strain E. coli CFT073, we performed the in-flow biofilm experiments to check the impact of specific and non-specific phages showing depolymerase activity. For this study, we used Ciprofloxacin antibiotic as a positive control and phage untreated control. Here, the quantification method used is based on cell staining with a LIVE/DEAD BacLight Bacterial Viability kit. Therefore, live cells are stained green, and dead cells or metabolically inactivated cells are stained red. In our study, Biomass measures represent the quantity of live cells after each treatment. In these types of experiments, Biomass reduction can be associated with increased dead cells and/or biofilm removal due to the detachment of cells after flow exposure. Our results agree with the second scenario because, in images, we can observe holes in affected biofilms due to a reduction in live cells (green). Still, no significative differences in dead cells (red) are observed in treated channels concerning control. Therefore, with this technique, we can see that biofilm reduction by phages with depolymerase activity is associated with matrix degradation more than cell lysis, thus, host specificity is not a requirement.
Section 3.7 In-vivo phage therapy in G. mellonella infection model:
“It was observed that out of the 2 phages used for the in vivo experiments, IBEC40 was cross-infective, whereas IBEC77 was not cross-infective with the E. coli CFT073 strain.”
- Earlier in the manuscript it is mentioned that IBEC40 was cross infective against MG1655 and not specific against CFT073 whereas IBEC77 was specific to CFT073. Please clarify.
Response: Thank you for your suggestion. IBEC40 is not specific to the E. coli CFT073 strain, but IBEC77 is. The clarification has been added at the beginning of this section.
- What concentration of ciprofloxacin was used?
Response: We appreciate the reviewer for this observation. The dose (20 mg/kg larva) is already detailed in the materials and methods, section 2.10. We have now added ciprofloxacin concentration in the legend of figure 6 for a better understanding of the figure.
- 1 dose of ciprofloxacin alone is preventing the larval death throughout the observation period indicating an efficient treatment intervention. However single or double dose of both of the specific or non-specific phages reduced survival of the larva. As ciprofloxacin alone itself is efficiently inhibiting larval death, combining the ciprofloxacin with any of the phages does not evaluate phage efficacy, hence the (antibiotic + phage) combination study is not relevant since the phage toxicity test is already been evaluated.
Response: We agree with the reviewer. Combining ciprofloxacin and the phages only reaffirm that we can still maintain 100% larva survival due to antibiotic efficiency. For further studies, we should test a dose of ciprofloxacin that could not inhibit total larva death and then see if the combined therapy improves survival.
-1X indicates one dose at 1 h post-infection (h.p.i) â–² also indicates one dose at 1 h post-infection (h.p.i).
Response: The reviewer is correct. However, in the combination of phages and antibiotics, the second dose was administered at a different time than when we used the second dose in phages individually (x2). Although the first dose has always been at 1 hour post-infection, a new symbology was added to avoid confusion between treatments.
- Conclusions
“Our work demonstrates that bacteriophages with a short latency period and high burst size, with depolymerized activity, can be the best alternative for treating uropathogenic and invasive biofilm-forming E. coli pathogens.”
- As any comparative study (phage with long latency, small burst size and without depolymerase activity) did not perform than how does this study can be concluded with the aforementioned statement.
Response: We have modified the statement as per suggestion and have provided clarification in the conclusion section of the article.
Suggestions:
- Bacterial Challenge test with different MOI must be included to evaluate the infectivity curve of the phages against bacteria.
- In-vitro static antibiofilm assay with specific /non-specific phages and phage antibiotic combination to correctly evaluate the anti-biofilm efficacy need to do.
Response: Thank you for the suggestions provided. The required necessary explanations have been provided previously, and the changes have been made in the article. We will use this information for our future work.

Reviewer 2 Report
The manuscript describes the isolation of 21 phages against three Ecoli strains.
The plaque and phage morphology, cross infectivity, growth curve and effect on biofilm and galleria was conducted.
The data is interesting but there are however few major comments that should clarify to make this a better read.
Major comments:
1. Please clarify if ethical documentation was in place before collection of samples from patients. It is clear that patient data was not required (that is fine in some studies) and that the samples were biological waste as stated on lines 99-103. However, waste or not, these are human biological resources hence ethics are required to be in place for this to be used for research. Please see Relevant material under the Human Tissue Act 2004 | Human Tissue Authority (hta.gov.uk) and MRC-0208212-Human-tissue-and-biological-samples-for-use-in-research.pdf (ukri.org). I am not sure what is tenable in the countries where the work was done but this needs to be looked into. For this manuscript to be published, it is important to provide report/approval or or an exemption by an ethical review committee board as prescribed via MDPI | Research and Publication Ethics and Declaration of Helsinki – WMA – The World Medical Association
2. Line 206-208, It is unclear if the cultures were refreshed in any way within these time periods?
3. Line 231-239, good idea to remove toxin from phage samples. However, this was not done for the bacterial culture. Would be helpful to remove toxins from the initial culture, washing and transfer to PBS to determine effect on larvae was as result of effect of colonization and not an outright killing from the toxin
Minor comments:
1. Italicise biological names. See lines 49, 50, 55, 57 and so many places across the manuscript.
2. Line 123, why was OD measurements not done? Judging by visual inspection would be very limited
3. Line 124, to what degree was lysis got to? And OD values would be more informative.
4. Line 173-175, how can you put 500 ml culture of E. coli in 50 ml tubes?
5. Line 208, how did you determine if biofilm was formed maximally? Were the biofilm characteristics same for all the strains? Ie were the biofilm profile same for all the strains. Provide this data as supplementary data.
6. Line 244, not clear, was phage dose also administered to the larvae at 5 h post infection? Clarify on figure 6
7. Table 1, would be more informative to have the pictures of plates showing the different morphology of the plagues stated on line 265-267 and 285-287, or simply how the differ in morphology. The table is not very informative to me for this data.
8. Figure 3, not convinced B) IBEC77 is a podovirus by the image. Looks more or less like a broken phage showing detached capsid. Please provide better image.
9. Figure 4 and 5, why was 72 h biofilm used for static data (figure 4) and 96 h for continuous data (figure 5)? Hard to compare the data and as they show different information. Would be nice to have data for static and continuous biofilm for the same time point for better comparison.
10. Figure 6,
a. What was the concentration of CPX used on galleria and how was this value determined?
b. Its not clear if the strains actually colonised the larvae or were simply killed them from toxins accumulated in the initial bacterial culture used to inoculate the larvae.
c. What was the LD50 for each of the strains?
d. Also line 404-408, I’m sure what was determined here as the survival data should be supplemented with either colonisation CFU counts or some biofilm assay to test non specificity of phage and depolymerase activity. At the end of the experiment, data for 1xIDEC77 treatment and the survival for no specific phage was the same for 2xIDEC77 and 1x/2x IDEC40
11. No comments made on CPX activity. It seems the antibiotics is the effective substance here and having therapeutic activity than the phages. Hence more clarification is needed on the impact of the antibiotic and synergistic effect of antibiotic/phage treatment on survival of the larvae.
Author Response
See enclosed file
Dear reviewer 2
Thank you for thoroughly reviewing our paper and for the opportunity to submit a revised version. We much appreciate the reviewer's constructive comments on our manuscript (Manuscript cells-2083777, Accessing the in vivo efficiency of clinically isolated phages against uropathogenic and invasive biofilm-forming Escherichia coli strains for phage therapy), which have been of great help and have improved the manuscript over the previous version. We are really thankful for your work on our manuscript.
Our responses to his/her comments are detailed below (in red).
With the manuscript changes detailed below and our answers to the reviewer's comments, we hope you will now find the revised version of our manuscript acceptable for publication in Cells.
Sincerely,
Dr. Eduard Torrents
Response to Reviewer 2:
The manuscript describes the isolation of 21 phages against three E. coli strains.
The plaque and phage morphology, cross infectivity, growth curve and effect on biofilm and galleria was conducted.
The data is interesting but there are however few major comments that should clarify to make this a better read.Major comments:
- Please clarify if ethical documentation was in place before collection of samples from patients. It is clear that patient data was not required (that is fine in some studies) and that the samples were biological waste as stated on lines 99-103. However, waste or not, these are human biological resources hence ethics are required to be in place for this to be used for research. Please see Relevant material under the Human Tissue Act 2004 | Human Tissue Authority (hta.gov.uk) and MRC-0208212-Human-tissue-and-biological-samples-for-use-in-research.pdf (ukri.org). I am not sure what is tenable in the countries where the work was done but this needs to be looked into. For this manuscript to be published, it is important to provide report/approval or or an exemption by an ethical review committee board as prescribed via MDPI | Research and Publication Ethics and Declaration of Helsinki – WMA – The World Medical Association
Response: Thank you for your suggestion. We have included a statement regarding the clinical sample sampling. “The clinical sputum samples from cystic fibrosis patients and urine pooled samples from patients with different, not defined urine infections were processed before phage isolation. Informed consent was not required because we collected a microbiology laboratory waste product, and all comorbidity data were identified, excluding potential patient risks”. In addition, we have included the ethically approved protocol (PR(AG)275/2019) to transfer clinical waste material samples to the IBEC laboratories.
- Line 206-208, It is unclear if the cultures were refreshed in any way within these time periods?
Response: Thanks for your comment. For every new experiment, fresh cultures were used unless they were incubated in control conditions for the generation of biofilms or biofilm eradication/removal experiments.
- Line 231-239, good idea to remove toxin from phage samples. However, this was not done for the bacterial culture. Would be helpful to remove toxins from the initial culture, washing and transfer to PBS to determine effect on larvae was as result of effect of colonization and not an outright killing from the toxin
Response: Thank you for this critical comment. For the in vivo studies, bacteria cultures were always washed with PBS. Please, see the new information added in materials and methods, section 2.10.
Minor comments:
- Italicise biological names. See lines 49, 50, 55, 57 and so many places across the manuscript.
Response: Thank you for the suggestion. We have made the required changes.
- Line 123, why was OD measurements not done? Judging by visual inspection would be very limited
Response: Thank you for the suggestion. We have made the required changes.
- Line 124, to what degree was lysis got to? And OD values would be more informative.
Response: Thank you for the suggestion. We did the OD check and have included the OD in the revised manuscript, but the cell density and phage lytic activity were affected by many factors, as well as the time of incubation which usually affected the reading; hence, we have now provided the OD to determine the effective lysis event.
- Line 173-175, how can you put 500 ml culture of E. coli in 50 ml tubes?
Response: Thank you for the correction. We have modified the statement. It was a typo error.
- Line 208, how did you determine if biofilm was formed maximally? Were the biofilm characteristics same for all the strains? Ie were the biofilm profile same for all the strains. Provide this data as supplementary data.
Response: The biofilm characteristics were different, and we have provided the data as supplementary data.
- Line 244, not clear, was phage dose also administered to the larvae at 5 h post infection? Clarify on figure 6
Response: We thank the reviewer for this observation. Yes, phages were tested with one dose (x1) and also with two doses x2 (1 h post infection and 5 h post infection). We have clarified the legend of figure 6.
- Table 1 would be more informative to have the pictures of plates showing the different morphology of the plagues stated on line 265-267 and 285-287, or simply how the differ in morphology. The table is not very informative to me for this data.
Response: Thanks for the suggestion. We have now provided the picture of plaque morphology in supplementary figures 1 and 2.
- Figure 3, not convinced B) IBEC77 is a podovirus by the image. Looks more or less like a broken phage showing detached capsid. Please provide better image.
Response: We have reconfirmed the image as per your suggestion. It is a podovirus. We are providing a reference for a similar study.
Baig, A., Colom, J., Barrow, P., Schouler, C., Moodley, A., Lavigne, R., & Atterbury, R. (2017). Biology and Genomics of an Historic Therapeutic Escherichia coli Bacteriophage Collection. Frontiers in microbiology, 8, 1652. https://doi.org/10.3389/fmicb.2017.01652
- Figure 4 and 5, why was 72 h biofilm used for static data (figure 4) and 96 h for continuous data (figure 5)? Hard to compare the data and as they show different information. Would be nice to have data for static and continuous biofilm for the same time point for better comparison.
Response: The protocol for static and in-flow biofilm experiments were designed to obtain complete mature and well-established E. coli biofilms. As biofilm formed under continuous flow is exposed to additional factors, like shear stress, they require more growth time. Flow conditions also impact biofilm characteristics itself, for example, in biofilm structure. Therefore, our intention, including the flow biofilm experiments, was not to compare with static biofilms directly. Instead, we used the static model as an initial evaluation technique. Then we moved to a more realistic in vitro model to test the antibiofilm effect of phage specificity and depolymerase activity in an E. coli strain.
- Figure 6,
- What was the concentration of CPX used on galleria and how was this value determined?
Response: We appreciate the reviewer's observation. The concentration of CPX (20 mg/kg larva) is described in the materials and methods; see section 2.10. This concentration was selected according to the literature and the fact that this dose is like the respective accepted dose in humans. We have now provided references that back up our selection. The concentration of CPX has also been added to the legend of figure 6.
- Its not clear if the strains actually colonised the larvae or were simply killed them from toxins accumulated in the initial bacterial culture used to inoculate the larvae.
Response: We thank the reviewer for making this comment. The bacterial culture of E. coli CFT073 was washed with PBS three times to remove the toxins that could contain. This information has been added to materials and methods. Please, see section 2.10, second paragraph.
- What was the LD50 for each of the strains?
Response: Thank you for this observation. In toxicology, the lethal dose measures the dose required to kill half the members of a tested population after a specified test duration. In our case, we are not testing different dosages. However, we do see differences in the time when 50 % of larvae have already died. For E. coli CFT073, this measurement is 22 hours. In the case of IBEC77 (x1), it is 38 hours, while for IBEC77 (x2), it is 50.5 hours. For IBEC40 (x1), it is 28.5 hours, while for IBEC40 (x2), it is 23. This information has been added to the results, section 3.7.
- Also line 404-408, I’m sure what was determined here as the survival data should be supplemented with either colonisation CFU counts or some biofilm assay to test non specificity of phage and depolymerase activity. At the end of the experiment, data for 1xIDEC77 treatment and the survival for no specific phage was the same for 2xIDEC77 and 1x/2x IDEC40
Response: We appreciate the reviewer for pointing this out. Yes, it is the same at the end, but IBEC77 (x1) takes more time than IBEC40 to cause the same mortality. A paragraph about the time it takes to have 50% larvae mortality has been added in section 3.7.
- No comments were made on CPX activity. It seems the antibiotics is the effective substance here and having therapeutic activity than the phages. Hence more clarification is needed on the impact of the antibiotic and synergistic effect of antibiotic/phage treatment on the survival of the larvae.
Response: We thank the reviewer for pointing this out. The antibiotic maintains 100% larva survival on its own, so the combination of treatments does not test the efficacy of phages. A few sentences have been added with more clarification at the end of section 3.7.

Reviewer 3 Report
The manuscript submitted by Sanmukh et al. report that clinically isolated phages such as IBEC48, IBEC77, and IBEC3 fight against biofilm-forming Escherichia coli strains for phage therapy. In my opinion, the manuscript could be published after some minor revisions.
1) Abstract: the background and introduction are too long, almost 2/3 of the abstract length. Please concisely describe the results and conclusions in detail.
2) Line 95: 0.1 OD = ? CFU/mL, could you please give the information?
3) Line 225: Pls correct “1 x 107 pfu/mL”.
4) Figure 1 and its legend need correct (there are two “B)”).
5)I did not quite clearly catch the significant difference between fig.4 and fig.5.
6) Biofilm protect bacterial cells from phage infections by forming a matrix barrier. I wonder how your phages break through the outside biofilm barrier and kill the inside Escherichia coli strains? This point deserves a deep discussion.
Author Response
See enclosed file
Dear Reviewer 3
Thank you for thoroughly reviewing our paper and for the opportunity to submit a revised version. We much appreciate the reviewer's constructive comments on our manuscript (Manuscript cells-2083777, Accessing the in vivo efficiency of clinically isolated phages against uropathogenic and invasive biofilm-forming Escherichia coli strains for phage therapy), which have been of great help and have improved the manuscript over the previous version. We are really thankful for your work on our manuscript.
Our responses to his/her comments are detailed below (in red).
With the manuscript changes detailed below and our answers to the reviewer's comments, we hope you will now find the revised version of our manuscript acceptable for publication in Cells.
Sincerely,
Dr. Eduard Torrents
Response to Reviewer 3:
The manuscript submitted by Sanmukh et al. report that clinically isolated phages such as IBEC48, IBEC77, and IBEC3 fight against biofilm-forming Escherichia coli strains for phage therapy. In my opinion, the manuscript could be published after some minor revisions
1) Abstract: the background and introduction are too long, almost 2/3 of the abstract length. Please concisely describe the results and conclusions in detail.
Response: Thank you for your comments. We have modified the abstract as suggested and included a concise description of the results obtained and conclusions.
2) Line 95: 0.1 OD = ? CFU/mL, could you please give the information?
Response: We have included the amount of initial E. coli cells in CFU/ml. Here, 0.1 OD corresponds to 1 x 108 CFU/ml.
3) Line 225: Pls correct “1 x 107 pfu/mL”.
Response: We have corrected the value.
4) Figure 1 and its legend need correct (there are two “B)”).
Response: We have corrected the figure.
5)I did not quite clearly catch the significant difference between fig.4 and fig.5.
Response: Thank you for your concern. Figure 4 shows the antibiofilm effect of bacteriophages on 72 h static biofilms (96 well plates at 37ºC) from their original host. In contrast, Figure 5 shows the antibiofilm effect of two bacteriophages with depolymerase activity regarding host specificity against 96 h biofilm formed under flow conditions.
6) Biofilm protect bacterial cells from phage infections by forming a matrix barrier. I wonder how your phages break through the outside biofilm barrier and kill the inside Escherichia coli strains? This point deserves a deep discussion.
Response: Thanks for pointing this out. Phage cannot disaggregate biofilm by altering its extracellular matrix structure. But the principle is comparable to the effect of antibiotics over biofilms. Phages, as antibiotics, destroy the outer layers of bacteria, creating holes in the extracellular biofilm matrix that allow the exposition of bacteria sensitive to the antibiotics and phages. See the new sentences added in the discussion section.

Reviewer 4 Report
This is an interesting study about the use of phage as anti-biofilm candidate.
well presented study;
One point to add:
you demonstrated that combination of phage and ciproflaxacin was more effective than phage alone in the G. mellonella infection model. You should have shown this synergism in the biofilm effect study (figure 5), as this effect may be due to the better antibiofilm activity with combination of phage and ciprofloxacin.
Author Response
See enclosed file.
Dear Reviewer 4
Thank you for thoroughly reviewing our paper and for the opportunity to submit a revised version. We much appreciate the reviewer's constructive comments on our manuscript (Manuscript cells-2083777, Accessing the in vivo efficiency of clinically isolated phages against uropathogenic and invasive biofilm-forming Escherichia coli strains for phage therapy), which have been of great help and have improved the manuscript over the previous version. We are really thankful for your work on our manuscript.
Our responses to his/her comments are detailed below (in red).
With the manuscript changes detailed below and our answers to the reviewer's comments, we hope you will now find the revised version of our manuscript acceptable for publication in Cells.
Sincerely,
Dr. Eduard Torrents
Response to Reviewer 4:
This is an interesting study about the use of phage as an anti-biofilm candidate.
well-presented study;
One point to add:
you demonstrated that a combination of phage and ciprofloxacin was more effective than phage alone in the G. mellonella infection model. You should have shown this synergism in the biofilm effect study (figure 5), as this effect may be due to the better antibiofilm activity with a combination of phage and ciprofloxacin.
Response: Thank you for your comments. Although the G. mellonella model can be adapted to study biofilm infections, the protocol employed in our study represents an acute E. coli infection. No protocols are established for a biofilm infection in G. mellonella, as the high doses of bacteria ultimately kill the larvae. Therefore, the synergism between the phage and the antibiotic is not explained, but an antibiofilm effect is observed.

Round 2
Reviewer 1 Report
The authors have addressed all the queries satisfactorily. I endorse the manuscript for publication